# Entomopathogenic Nematodes for Field Control of Onion Maggot (*Delia antiqua*) and Compatibility with Seed Treatments

**DOI:** 10.3390/insects14070623

**Published:** 2023-07-11

**Authors:** Camila C. Filgueiras, Elson J. Shields, Brian A. Nault, Denis S. Willett

**Affiliations:** 1Department of Biology, University of North Carolina Asheville, One University Heights, Asheville, NC 28804, USA; camila@unca.edu; 2Department of Entomology, Cornell University, 2126 Comstock Hall, Ithaca, NY 14853, USA; 3Department of Entolomogy, Cornell University, Cornell AgriTech, 15 Castle Creek Drive, Geneva, NY 14456, USA; 4North Carolina Institute for Climate Studies, North Carolina State University, 151 Patton Avenue, Asheville, NC 28801, USA

**Keywords:** biological control, conventional management, soil pests, bulb yield

## Abstract

**Simple Summary:**

Onion maggot (*Delia antiqua*) can be a devastating pest of onion and related crops around the world. While management of this pest typically relies on insecticides to control these pests, growing resistance and desire for alternative control strategies has motivated a search for new methods to manage them effectively. We investigated the potential of using tiny beneficial roundworms called entomopathogenic nematodes to control onion maggots. We conducted surveys in onion-growing areas to see if these nematodes were naturally present, and also tested their compatibility with commonly used insecticides. Our field trials demonstrated that applying entomopathogenic nematodes can significantly reduce the number of onion plants destroyed by onion maggots, leading to higher crop yields. These nematodes could be a useful tool for onion farmers, whether they practice conventional or organic farming. When combined with other strategies like insecticide seed treatments, the use of entomopathogenic nematodes can provide effective control of onion maggots while boosting crop productivity.

**Abstract:**

Onion maggot (*Delia antiqua*) is a prominent pest of allium crops in temperate zones worldwide. Management of this pest relies on prophylactic insecticide applications at planting that target the first generation. Because effective options are limited, growers are interested in novel tactics such as deployment of entomopathogenic nematodes. We surveyed muck soils where onions are typically grown to determine if entomopathogenic nematode species were present, and then evaluated the compatibility of entomopathogenic nematode species with the insecticides commonly used to manage *D. antiqua*. We also evaluated the efficacy of these entomopathogenic nematodes for reducing *D. antiqua* infestations in the field. No endemic entomopathogenic nematodes were detected in surveys of muck fields in New York. Compatibility assays indicated that, although insecticides such as spinosad and, to some extent, cyromazine did cause mortality of entomopathogenic nematodes, these insecticides did not affect infectivity of the entomopathogenic nematodes. Field trials indicated that applications of entomopathogenic nematodes can reduce the percentage of onion plants killed by *D. antiqua* from 6% to 30%. Entomopathogenic nematodes reduced *D. antiqua* damage and increased end of season yield over two field seasons. Applications of entomopathogenic nematodes may be a viable option for reducing *D. antiqua* populations in conventional and organic systems. Together with other management tactics, like insecticide seed treatments, applications of entomopathogenic nematodes can provide a yield boost and a commercially acceptable level of *D. antiqua* control.

## 1. Introduction

*Delia antiqua* (Meigen) (Diptera: Anthomyiidae) is one of the most important insect pests of onion (*Allium cepa* L.) worldwide [1,2]. In addition, *D. antiqua* attacks garlic (*Allium sativum* L.), scallions (*Allium fistulosum* L.) and chives (*Allium schoenoprasum* L.) [3,4,5]. Because crop losses from *D. antiqua* can approach 100% without management, most commercial production in temperate regions of the Americas, Europe, and Asia depend on intensive management of this pest [1,2]. Despite this intensive management, control remains a challenge especially in areas where crop rotation is not practiced, where resistance has developed, or where long-used pesticides such as chlorpyrifos are being discontinued [2,6].

Commercial management of *D. antiqua* often involves prophylatic insecticide applications at planting via seed treatments and in-furrow drenches [1]. These treatments primarily target the first generation of maggots. In temperate regions of North America, *D. antiqua* has three generations per year and the first generation emerges from overwintering pupae in late April and May [7,8,9,10]. Larvae from the first generation feed on belowground portions of the onion plant, which either kills the plant or inflicts damage that makes it susceptible to pathogen infection [1,11]. Feeding damage by the second- and third-generation *D. antiqua* may not kill the plant, but can render the bulbs unmarketable.

Cyromazine and spinosad are two of the most effective insecticides for managing *D. antiqua* [1,2]. Both are delivered as seed treatments, but are available in separate seed treatment packages that also include thiamethoxam plus three fungicides, mefenoxam, fludioxonil and azoxystrobin [12,13]. Until recently, chlorpyrifos was commonly used as a drench treatment at planting either alone or with insecticide seed treatments for *D. antiqua* control. However, chlorpyrifos was banned for use on all crops by the EPA in early 2022. These insecticides tend to only provide control of first-generation *D. antiqua*, while no insecticides are used to target subsequent generations. Onion growers are interested in non-chemical tactics for managing *D. antiqua*, especially options that may manage all three generations.

A promising alternative to insecticides that may have an impact on reducing multiple *D. antiqua* generations is entomopathogenic nematodes. Entomopathogenic nematodes are tiny roundworms whose infective juvenile stage seeks out and infects and kills insects [14]. Control of insect pests using entomopathogenic nematodes as biological control agents has proven successful against some insect pests [15] and has been evaluated in relation to control of other *Delia* pests. While entomopathogenic nematodes have shown promise in laboratory trials against *Delia* species such as *D. antiqua*, *D. platura*, and *D. radicum*, field efficacy either was not effective (in the case of *D. radicum* [16,17]) or not evaluated (in the case of *D. antiqua* [18] and *D. platura* [19]).

Entomopathogenic nematodes have a number of potential advantages for control of *Delia* pests, particularly *D. antiqua*. After initial introduction, entomopathogenic nematodes can potentially persist and engender long-term control [20]. This attribute is especially appealing in New York where onion fields are rarely rotated and *D. antiqua* populations can persist at high levels.

In developing entomopathogenic nematodes as biological control agents, species considerations are important. Selecting a virulent entomopathogenic nematode strain that can survive in field conditions is critical. Equally important is making sure that these biological control agents are compatible with current control methods, like soil-applied insecticides. Pesticides used in conjunction with nematodes can have vastly different effects depending on the interaction between the pesticide and the entomopathogenic species used. In some cases, the use of pesticides in conjunction with entomopathogenic nematodes can result in synergistic effects for pest control [21]. In others, pesticides can kill nematodes likely resulting in decreased efficacy [22,23]. In other cases, no synergistic effects are observed [18,24].

To explore the potential integration of biological control into an IPM program for *D. antiqua* in onion, our objectives were to (1) determine if endemic entomopathogenic nematodes are present in muck soils in upstate New York where onions are grown continuously, (2) explore the impact of insecticides commonly used for *D. antiqua* control on entomopathogenic nematode health and infectivity, (3) to determine if applications of entomopathogenic nematodes in onion fields can reduce onion plant stand loss caused by *D. antiqua*, and (4) if these entomopathogenic nematodes can persist in these fields for multiple years.

## 2. Materials and Methods

### 2.1. Soil Surveys

To determine if entomopathogenic nematodes occur in muck soils in upstate New York and to examine whether entomopathogenic nematodes were present in our field trial locations, soil cores were collected and then placed in cups containing *Galleria mellonella* larvae, which were used as hosts for entomopathogenic nematodes.

To evaluate the presence of existing populations of entomopathogenic nematodes in muck soils of New York, 12 onion fields on muck soils were sampled across five counties (Genesee, Yates, Steuben, Wayne, and Oswego) in central and western New York. Each of these sites were chosen because of the the presence of muck soils and a history of continuous onion production for multiple years. GPS coordinates can be found in Appendix A Table A1. At each location, 10 soil cores were taken along a 100 m transect in the middle of each field. Cores were taken such that the top 20 cm of soil (in a 2 cm diameter core) was removed and placed into 250 mL containers to which 10 late instar *G. mellonella* larvae were added and left under controlled (25 °C, no light) conditions for seven days. After seven days, *G. mellonella* larvae were removed, placed on white traps and monitored both for symptomology of entomopathogenic nematode infection and entomopathogenic nematode emergence [25]. Symptomatic *G. mellonella* larvae were dissected to confirm presence or absence of entomopathogenic nematodes.

### 2.2. Compatibility Trials

To determine if entomopathogenic nematodes were compatible with insecticides commonly used as onion seed treatments used to control *D. antiqua*, mortality of entomopathogenic nematode species was assessed after exposure to Entrust SC (a.i. spinosad 22.5%—EPA Reg. No. 62719-621) and Trigard 75WP (a.i. cyromazine 75%—EPA Reg. No. 100-654). To do so, cohorts of entomopathogenic nematode species that had demonstrated efficacy in killing *D. antiqua* in previous trials were placed in vials containing the insecticide solution and their mortality and ability to infect evaluated after 48hrs.

#### 2.2.1. Preparation

Specifically, a 2 mL solution was prepared containing insecticide treatments and nematode species, with all nematode-pesticide combinations evaluated with 5 replications. Treatments included spinosad, cyromazine, and a water-only control. Entomopathogenic nematode species consisted of *Heterorhabditis bacteriophora*, *Steinernema diaprepesi*, *S. feltiae*, and *S. khuongi*. *H. bacteriophora* and *S. feltiae* strains were isolated from upstate NY by the Shields lab. *S. diaprepesi* and *S. khuongi* strains were isolated from Florida by the LW Duncan lab at University of Florida. All nematode species were subsequently confirmed via barcoding using established techniques [26]. Solutions in vials contained 2000 one-week old infective juveniles (IJs) (2 mL total volume or 1000 IJs/mL) or none in water-only controls. Insecticide dosage was calculated as double the concentration recommended in the field (8.76 mL/L and 3.72 g/L, for spinosad and cyromazine respectively). We chose these amounts on the off chance that nematodes would come in direct contact with these solutions. On a per cohort basis, these concentrations result in nematodes cohorts being exposed to either 3.9 mg of spinosad or 5.58 mg cyromazine in 2 mL of water. These amounts are orders of magnitude above what would be encountered as seed treatments (0.2 mg spinosad per seed and 0.225 mg cyromazine per seed) in the field. Pesticide stock solutions contained Tween 20; water only controls also contained Tween 20 at the same concentration.

#### 2.2.2. Mortality

Entomopathogenic nematode infective juvenile cohorts were left in solution for 48 h in the dark at 25 ± 1 °C. Each cohort (with 5 cohorts each per treatment as described above) was a replication resulting in five replications for each combination of nematode species and treatment. After 48 h, nematodes were triple washed with distilled and deionized water over a fine (325 mesh) screen to remove the pesticide solution and then re-suspended in 2 mL of distilled and deionized water. Three aliquots of 0.1 mL each were placed under a stereoscope for determining mortality of the entomopathogenic nematodes for each replication. Entomopathogenic nematodes were considered dead if they were straight and needle-like under the stereoscope and did not respond to prodding with the tip of a needle.

#### 2.2.3. Infectivity

Immediately following assessments of mortality, 1 mL of the washed and resuspended nematode solution at 1000 IJs/mL was applied to a petri dish (55 mm diameter) containing filter paper and 5 *G. mellonella* larvae. This was done for each cohort described above resulting in five replications for each combination of nematode species and treatment. These dishes containing entomopathogenic nematode inoculated larvae were left in the dark for 6 days at 25 ± 1 °C. The seventh day after inoculation, any dead larvae were dissected and the presence or absence of entomopathogenic nematodes confirmed under a stereoscope.

### 2.3. Field Trials

To determine if entomopathogenic nematodes could be used to reduce *D. antiqua* populations and their damage in the field, entomopathogenic nematodes were applied to commercial onion fields on muck soils in Oswego, NY in small plot trials in 2019 and larger scale trials in 2020. Onion plants killed by *D. antiqua* were evaluated and recorded throughout both seasons using established methodology [1,12].

The presence of entomopathogenic nematodes in the soil at the location of our *D. antiqua* control trials were sampled within the onion-growing seasons in 2019, 2020, and 2021. Soil cores were taken as above from entomopathogenic nematode-treated and untreated control plots prior to each of three entomopathogenic nematode applications and in the season following applications. This assessment allowed us to determine if: (1) Entomopathogenic nematodes were present in the soil prior to the initiation of our field trial; (2) Entomopathogenic nematodes from previous applications would persist in the soil; and (3) untreated control plots had become contaminated with entomopathogenic nematodes from movement through the soil or transfer on farm implements. Following collection of soil cores, entomopathogenic nematode presence was evaluated as described above using *G. mellonella* larvae to screen for entomopathogenic nematode infection.

#### 2.3.1. 2019 Field Trials

This experiment was conducted on muck soil on a commercial farm south of Oswego, New York (GPS: 43°27′04.3″ N 76°23′58.6″ W) in 2019. All experimental plots were planted on 16 May with onion seeds *A. cepa* L.; var. ‘Fortress’ treated with the fungicides penflufen (EverGol Prime) and thiram (Thiram) to protect seedlings against onion smut (*Urocystis colchici*) and *Rhizoctonia* spp., respectively. No insecticides were applied at planting. Plots were two rows (approximately 40 cm apart) by 3.5 m long with a 3 m unplanted buffer surrounding plots. Each plot had approximately 100 plants (see data citation for exact numbers). There were two treatments, entomopathogenic nematode treated and untreated, that were replicated four times and arranged in a randomized complete block design. Onions were maintained by the grower following crop production practices that do not impact *D. antiqua*.

Onion plants killed by *D. antiqua* were recorded weekly or every other week in plots until near harvest. The cumulative percentages of onion plants killed by *D. antiqua* in entomopathogenic nematode-treated plots were compared with those in control plots receiving no application of entomopathogenic nematodes.

Entomopathogenic nematode infective juveniles were applied at a rate of approximately 150 million per acre on 23 May, 6 June, and 1 July. *H. bacteriophora* and *S. feltiae* infective juveniles in a 50:50 ratio (each at 1000 IJs/mL) were applied via backpack sprayer over onion rows using 1 L total solution per plot (500 mL/row). The same amount of water (without nematodes) was applied to control plots at the same time. These entomopathogenic nematode species were chosen because of their hardiness to conditions in upstate New York, their complementary stratified preference for soil depths, and their ability to maintain infectivity in the presence of spinosad and cyromazine.

#### 2.3.2. 2020 Field Trials

In 2020, in a separate location from the 2019 field trials, onion plant stands in entomopathogenic nematode treated plots were compared with control plots receiving no application of entomopathogenic nematodes with four replications in randomized control block design. All experimental plots were planted in late April with onion seeds *A. cepa* L.; var. “Highlander”. Plots were located in muck soils in Oswego County (GPS: 43°27′04.3″ N 76°23′58.6″ W) and were 4 rows (each approximately 40 cm apart) by 30 m with a 5 m buffer between plots on all sides with approximately 1700 plants per plot.

Entomopathogenic nematode infective juveniles were applied at a rate of approximately 1 billion per acre on 20 April, 16 June, and 6 August (beginning, middle, and end of the season respectively). To do so, *H. bacteriophora* and *S. feltiae* infective juveniles in a 50:50 ratio (each at 1000 IJs/mL) were applied via backpack sprayer over onion rows using 4 L total solution per plot (1 L/row). The same amount of water (without nematodes) was applied to control plots at the same time. The higher rate in 2020 was selected because of the larger nature of the field trial and because of co-occurrence of seed treatments and conventional *D. antiqua* management.

During and immediately following the first application, onion stands were evaluated for signs of onion maggot damage (by examining plants for the presence of *D. antiqua* larvae [12]) and stand counts taken (beginning of season stand count averaged approximately 1700 plants). All plants in this trial were managed under the same conventional approach as the rest of the field including use of the seed treatment package, FarMore F1500 (Syngenta, Greensboro, NC, USA) (a.i.’s including spinosad, thiamethoxam, mefenoxam, fludioxonil, and azoxystrobin), which is typically effective in managing *D. antiqua* infestations. However, FarMore FI500 has not provided a commercially acceptable level of control at this site [12].

### 2.4. Analysis

Data were collated in tabular form (comma separated values) then imported to R V4.2.0 using RStudio as an IDE [27,28]. The Tidyverse package was used to facilitate analysis and plotting [29]. The car, emmeans, and lmtest packages were use to assist in model development, analysis, and interpretation [30,31,32].

#### 2.4.1. Soil Surveys

Entomopathogenic nematode presence and absence data from soil surveys are reported as is with no additional analysis applied.

#### 2.4.2. Compatibility Trials

Linear mixed effects models were used to evaluate the fixed effects of species, pesticide, and their interaction on nematode infective juvenile mortality. Aliquot was included as a random effect to account for repeated measures. Best fit models were chosen based on visual inspection of model diagnostic plots for adherence to model assumptions, analysis of deviance, Akaike and Bayesian information criteria, and pseudo-R^2^ values. Post-hoc comparisons were made using Dunnett’s test to compare treatment to controls and correct for the family-wise error rate.

Logistic regression was used to evaluate effect of species, pesticide treatment, and their interaction on the ability of exposed cohorts of nematodes to induce mortality in *G. mellonella* larvae. Best fit models were chosen based on visual inspection of model diagnostic plots for adherence to model assumptions, analysis of deviance, Akaike and Bayesian information criteria, likelihood ratio tests, and pseudo-R^2^ values.

#### 2.4.3. Field Trials

Linear mixed effects models were used to evaluate the fixed effects of entomopathogenic nematode treatment on onion loss due to onion maggot for each field season with date and block included as random effects to account for repeated measures and spatial heterogeneity. Best fit models were selected as above and post-hoc comparisons made with Tukey’s HSD test.

## 3. Results

### 3.1. Soil Surveys

Surveys for existing entomopathogenic nematodes at onion muck sites across western New York did not reveal any extant endemic populations at the time of sampling. In the locations where the **Impact of Entomopathogenic Nematodes on *D. antiqua* in Onion Field Trials** were conducted, *H. bacteriophora* and *S. feltiae* entomopathogenic nematodes were recovered months and a year following application. Following the first three applications in 2019, entomopathogenic nematodes were recovered from sites of application in 2019 and at the beginning of the 2020 field season (Figure 1). No entomopathogenic nematodes were ever recovered in control (non entomopathogenic nematode treated plots). Entomopathogenic nematodes were not recovered in 2021 at the location where entomopathogenic nematodes were applied in 2020 (Figure 1).

### 3.2. Compatibility Trials

#### 3.2.1. Mortality

Exposure to commonly used pesticides had a significant ($\chi^{2}$ = 971.25, df = 2, p<0.01) effect on entomopathogenic nematode mortality, as did the species exposed ($\chi^{2}$ = 1000.14, df = 4, p<0.01) and the interaction between species and treatment ($\chi^{2}$ = 698.66, df = 8, p<0.01) (Figure 2). Exposure to spinosad significantly (t>20, df=206, p<<0.01) increased mortality compared with controls. Exposure to cyromazine either had no significant effect for *H. bacteriophora*, *S. feltiae*, and *S. khuongi WEB* or significantly decreased mortality for *S. diaprepesi* (t=−13, df=206, p<<0.01) and *S. khuongi ARCA* (t=−2.8, df=206, p=0.01).

#### 3.2.2. Infectivity

Of those nematodes that survived exposure to pesticides, exposure to pesticides had very little effect on the ability of entomopathogenic nematodes to kill *G. mellonella* larvae ($\chi^{2}$ = 0.05, df = 2, p<0.97) (Figure 3). Species ($\chi^{2}$ = 64.76, df = 4, p<0.0001) and the interaction between species and treatment ($\chi^{2}$ = 76.52, df = 8, p<0.0001) did significantly explain observed infectivity. Only one species of entomopathogenic nematode (*S. khuongi* ARCA) had an odds ratio significantly different from one (p<0.0001) suggesting that exposure to spinosad decreased their ability to infect their insect hosts. Other species were impacted as well, but just not significantly (p>0.05).

### 3.3. Field Trials

In both the 2019 and 2020 field season, applications of entomopathogenic nematodes reduced onion plant loss and increased end of season plant stands (in terms of bulb yield) (Figure 4).

In small plot field trials in 2019, the effect of entomopathogenic nematode applications was almost significant with the inclusion of all four replicates ($\chi^{2}$ = 3.19, df = 1, p=0.07). One replicate was flooded early in the season proximal to the time of the first nematode application. In treating this replication as an outlier and removing it from the analysis, the effect of entomopathogenic application becomes highly significant ($\chi^{2}$ = 12.11, df = 1, p=0.0005) and the estimated effect of entomopathogenic nematode application is a 52.9 ± 15% increase in end of season yield (t=3.42, df=34, p=0.002). Doing the same analysis with inclusion of the outlier, the estimated effect of entomopathogenic nematode application is a 30.3 ± 17% increase in end of season yield (t=1.8, df=47.0, p=0.08).

In large plot field trials in 2020, the effect of entomopathogenic nematode application was significant ($\chi^{2}$ = 21.7, df = 1, p<0.0001). Application of entomopathogenic nematodes increased end of season yield by 5.8 ± 1.2% (t=4.6, df=54.0, p<0.0001).

## 4. Discussion

Applications of entomopathogenic nematodes can reduce *D. antiqua* damage and increase end of season plant stands. The application of entomopathogenic nematodes resulted in a increased number of plants at the end of season compared to treatments without entomopathogenic nematodes. In the presence of high *D. antiqua* pressure and without insecticides at planting in 2019, treatments of entomopathogenic nematodes increased plant stands at the end of the season by approximately 30%. In the 2020 field season in larger plots with conventional insecticide seed treatment and with lower pest populations, applications of entomopathogenic nematodes continued to have a significant effect in reducing pest damage and increased plant stands by 6%. While the level of *D. antiqua* control provided by entomopathogenic nematodes is not sufficient to replace insecticide seed treatments, our results suggest that entomopathogenic nematodes could be used to enhance *D. antiqua* management above and beyond what is currently provided by standard insecticide seed treatments. This would be especially true in onion fields where control of *D. antiqua* using standard seed treatment packages has not been commercially acceptable (>5–10% plants killed) [33]. Even in onion fields where standard seed treatments are performing relatively well, very small increments in yield in a high-value crop like onion can increase profits considerably.

In the 2020 field season in larger plots with conventional insecticide seed treatment and with lower pest populations, applications of entomopathogenic nematodes continued to have a significant effect in reducing pest damage and increasing plant stands. These results suggest that entomopathogenic nematodes could be used to enhance *D. antiqua* control above and beyond what is currently used by growers.

Important to note is that in both field seasons, conventional management continued without alteration; pesticide sprays, fertilizer treatments, herbicide applications were all made by the grower on a regular basis in line with commonly recommended onion production practices in upstate New York. While many factors can cause EPN mortality and reduce efficacy post-application [34], the ability of entomopathogenic nematodes to achieve control in this environment suggests that these practices did not have a large negative impact on entomopathogenic nematode performance. However, future research is needed to assess this question.

Compatibility assays with compounds commonly used in seed treatments and potentially lethal to entomopathogenic nematodes also suggest that entomopathogenic nematodes could be a viable and effective control option in field environments. Spinosad caused high mortality in all the entomopathogenic nematode species tested. Cyromazine did not seem to affect mortality to the same extent, similar to results from other work [18]. Despite high mortality in some cases approaching 100%, exposure to these compounds did not for the most part impact the ability of entomopathogenic nematodes to infect their insect hosts. In other words, those infective juveniles that survived (even if it was just 1% of a few thousand nematodes) were able to still cause the same risk of mortality in their insect hosts. This ability suggests that, even if some entomopathogenic nematodes are killed by conventional management strategies such as commonly used seed treatments, the survivors are equally able to infect and kill insect pests. Other studies have highlighted the compatibility between entomopathogenic nematodes and commonly used pesticides reinforcing the viability of this combined approach [24,35].

Another factor pointing to the viability of entomopathogenic nematodes as an effective management tool for *D. antiqua* is their ability to potentially persist in the field. Surveys of onion muck fields suggested an absence of endemic biological control activity from entomopathogenic nematodes. Inoculating entomopathogenic nematodes into fields from which we were originally unable to detect entomopathogenic nematode activity resulted in establishing a population that we were able to detect a year after first inoculation following harvest, winter, and soil prep for a subsequent season albeit at low numbers. These results suggest that entomopathogenic nematodes could be able to persist in muck soils after initial applications. That being said, recovery of the original population was low in terms of the number of samples from which we were able to detect entomopathogenic nematode populations. The reasons why populations declined over seasons could be due to abiotic factors including weather and frequent disturbance of the soil via tillage and other soil disturbance practices (i.e., planting and harvest). While populations may persist, they may do so at numbers providing inadequate levels of control. If this were the case, additional applications of entomopathogenic nematodes may be needed to enhance the level of control. Moreover, persistence of entomopathogenic nematodes in non-rotated muck fields where onions are grown continuously for decades may over time substantially reduce *D. antiqua* populations.

The ability of entomopathogenic nematodes to survive in muck soils and be compatible with conventional seed treatments suggests that they could be an option for enhancing existing levels of *D. antiqua* control in both conventional and organic systems. Their ability to improve yield in conjunction with conventional management suggests that they could be an effective control tactic to be included in an integrated pest management approach to control *D. antiqua*.

## Figures and Tables

**Figure 1 insects-14-00623-f001:**
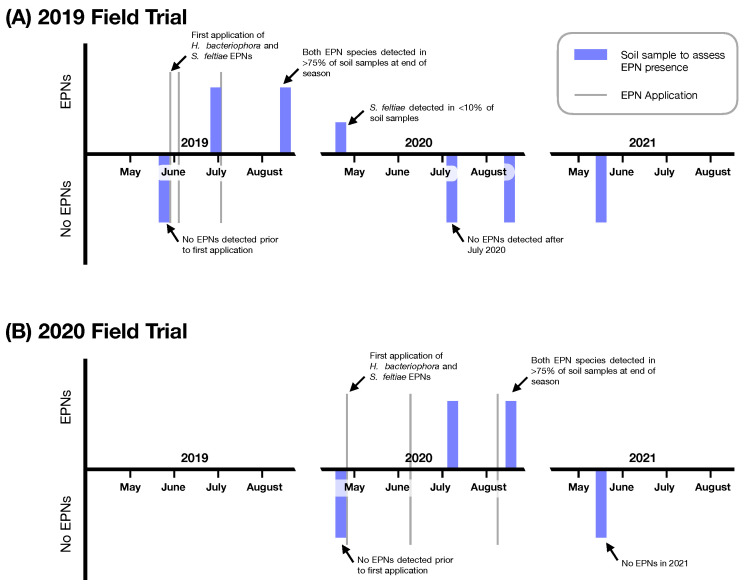
Entomopathogenic nematodes (EPNs) detected from soil samples at 2019 and 2020 Field sites. (**A**) Results of surveys at the 2019 Field Trials location. (**B**) Results of surveys at the 2020 Field Trial Location (separate and independent location from the 2019 Field Trials). Blue bars indicate timing of soil surveys for entomopathogenic nematodes; bars above the horizontal indicate detection of entomopathogenic nematodes and bars below the horizontal indicate no entomopathogenic nematodes detected. Vertical grey lines indicate timing of application.

**Figure 2 insects-14-00623-f002:**
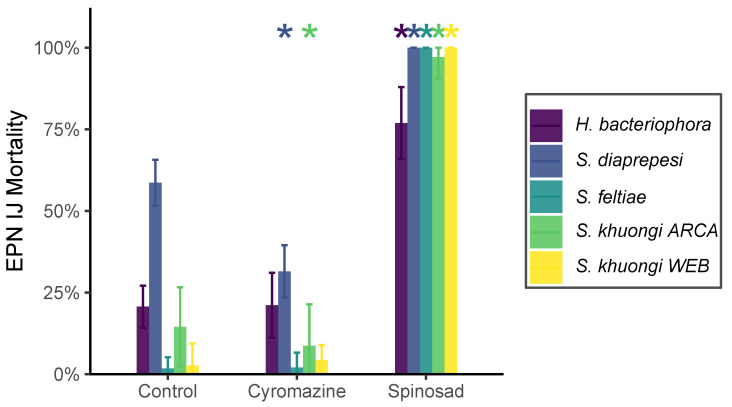
Mortality of entomopathogenic nematode cohorts of different species following exposure to solutions containing no pesticide (Control), spinosad, and cyromazine. Bar height and error bars denote mean mortality and standard deviation, respectively. Asterisks indicate a significant (*p* < 0.05) difference in mortality as compared to controls on a per-species basis.

**Figure 3 insects-14-00623-f003:**
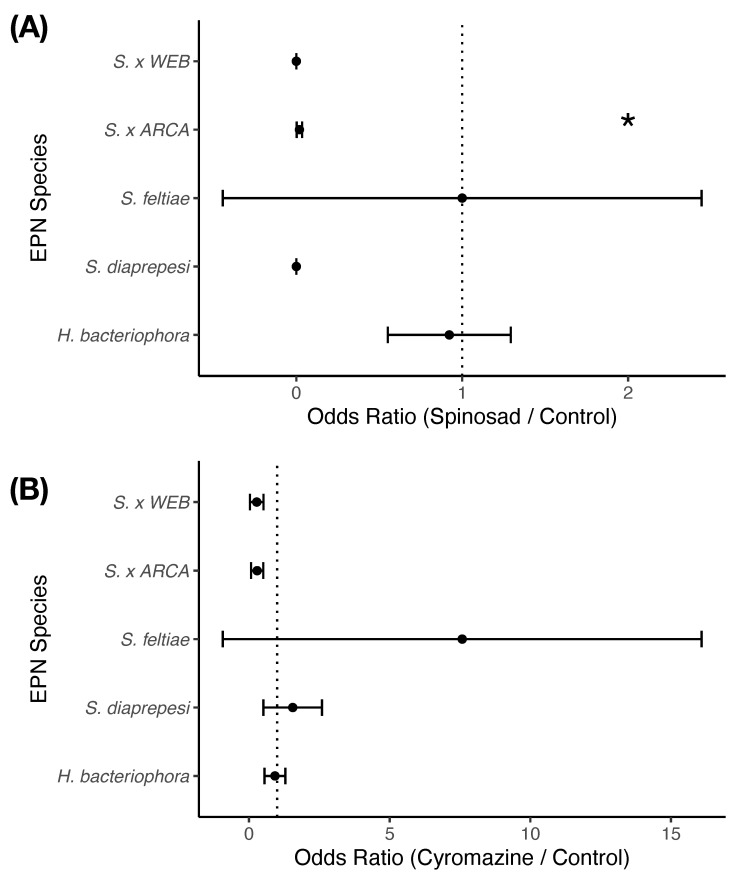
Infectivity of entomopathogenic nematode cohorts previously exposed to either (**A**) spinosad or (**B**) cyromazine. Odds ratios measure the change in infective potential of nematode cohorts after exposure to compounds as compared to controls. An odds ratio of one denotes no change; an odds ratio of less than one denotes a decreased ability of entomopathogenic nematodes to infect insects. Points and error bars denote mean odds ratio and standard error respectively. Asterisk indicates that the odds ratio is significantly different from one at *p* < 0.05.

**Figure 4 insects-14-00623-f004:**
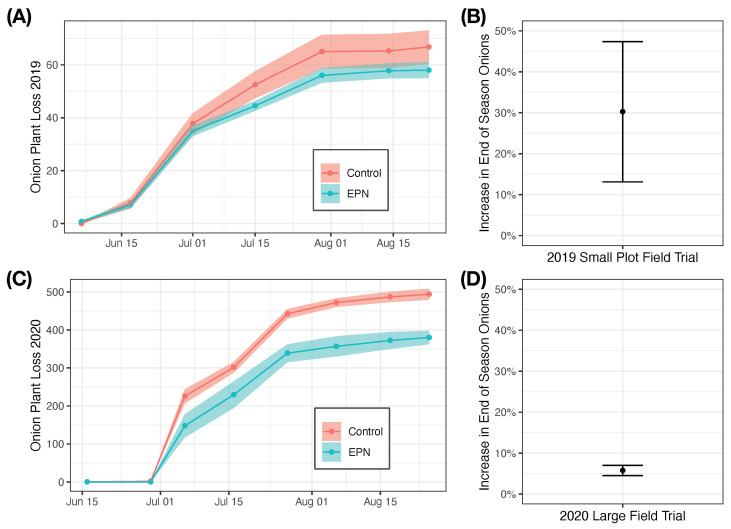
The effect of adding entomopathogenic nematodes on onion maggot damage. In 2019, in small plot trials without seed treatment, applications of entomopathogenic nematodes reduced cumulative onion loss (**A**) and resulted in an increase in end of season plant stand (**B**). In 2020, in larger plot trials with seed treatment, applications of entomopathogenic nematodes also reduced cumulative onion loss (**C**) and also resulted in an increase in end of season plant stand (**D**). In (**A**,**C**), solid lines and shaded region denote mean loss and standard error, respectively. In (**B**,**D**), points and error bars denote mean and standard error, respectively.

## Data Availability

Upon acceptance data and analysis code will be made available by request via a GitHub repository.

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
