# Peer review of "Entomopathogenic Nematodes for Field Control of Onion Maggot (Delia antiqua) and Compatibility with Seed Treatments"

_insects, 2023, doi:10.3390/insects14070623_

Round 1

Reviewer 1 Report

Important research demonstrating the potential of biological control of soil pests.

There are only two comments, in the results item.

Reviewer 2 Report

The authors declare that this manucript is intended to explore the potential integration of biological control into an IPM program for D. antiqua in onion but this manuscript has little to do with IPM since the basic IPM principles are missing.

The first step in IPM involves the implementation of a series of measures (prevention), such as crop rotation, that will create the conditions for a reduction in the risk of pest outbreaks and thus in the need for plant-protection measures.  It seems that your experimental conditions (you have to explain this specififically in M&M) were continuous onion cultivation (the opposite of IPM implementation, although in introduction you state “Despite this intensive management, control remains a challenge especially in areas where crop rotation is not practiced” (L25-26).

PREVENTION includes other structural strategies to keep harmful organism populations low (reduction of pest control need probability) such as other agronomic strategies (e.g. timing of sowing) and increasing agroecosystem complexity/stability starting with appropriate rotations, scientific production suggests that the more complex the ecosystem, the more stable it is; in addition we have to deal with several ecosystems because of plot variability (soil type, landscape, ..); continuos crop cultivation and continuous application of insecticides cause opposite conditions. 

Once these conditions have been created, before any decision on pest control is taken, harmful organisms must be monitored with adequate methods and tools, where available; tools should include observations in the field as well as scientifically sound warning, forecasting and early diagnosis systems; crops may only be treated when and where the assessment has found that levels exceed set economic thresholds; prophylactic use of insecticides is strongly against IPM but the main field trial is carried in a field where the seed were coated with insecticides; crop protection was based on a prophylactic approach following usual farmers’ practices.

The application of EPN cannot be considered an IPM approach in the described scenario!!!

Despite the remarks above, I recommend  the manuscript for publication, but after a revision that should solve the manuscript’s shortcomings including Incomplete Material and Methods.  

Detailed remarks and suggestions are reported below.

Abstract:

L 3-4:  I suggest to add hat scientific community should study and propose to the farmers an holistici effective IPM approach;

Key words:

Integrated Pest management has to be deleted if you do not modify the text;

Introduction:

Please describe the  most suitable IPM approach that growers should follow; you should cite at least a paper reporting about IPM principles and approaches; 

“Equally important is making sure that these biological control agents are compatible with current control methods, like soil-applied insecticides”: hardly, you can say that this is an IPM approach; one of the IPM principles states that biological alternatives should replace chemical pesticides.

Materials and methods All the actual agronomic information should be given: which were the previous crops of the fields? Continuous onion cultivation as stated in the introduction?

EPN are very suscepotible to soil moisture levels; the soil moisture pattern over the experimental patterns should be given and discussed;

The climatic parameters during  the experiment period have to be reported, at least temperatures  and rainfall (average, max, min).

L 171-174: how did you establish that the killed plant had been damaged by D. maggot?  

Did you extract nematodes from D. antiqua larvae?

Did you assess  D. antiqua larval instars actually affected by EPN?

 Results  

3.3 please, make clear which parameter you refer to ….. (you might add in terms of …)

Figure 4:  which measure unit? (n)?

 Discussion/Conclusions

The exploitation of EPN in a true  holistic IPM approach should be discussed.

Reviewer 3 Report

The authors in this study intend to assess the efficacy of entomopathogenic nematodes (EPNs) and their combination potential to control onion maggots. The EPNs and two insecticides spinosad and cyromazine were combined in two year field trials. The results indicated no negative impact of insecticides on infectivity of EPNs – there was a reduction in damage to onion plants and enhanced yield was noted. This ms is with in scope of journal – the manuscript presents interesting results and the authors generated good data.

This study is not well-designed and I have some reservations in Methodology section which is very confusing and need more clarity. Infact, in different assays the conclusions are based on single, un-repeated experiments and this is not acceptable. Replicates within an experiment measure variability within the system, whereas repetition of the experiments ensure repeatability of the results and lack of artefacts. Would you have the same conclusions if the study was repeated is the question that must be addressed?

The authors did not give relevant citations to most of the protocols (sub-sections 2.1 to 2.3) in Methodology section and overall this section is weakly written/explained. There is need to substantially improve this section with most relevant/authentic citations. There are some major flaws like the time of neither material collections nor experiments is given in some sub-sections of Methodology section, the authors must provide the time for each collection and experiment, how the assays were conducted etc., should be given in more detail and supported with relevant citation/s?

This article will provide more insights in commercial biological control of onion maggots so in my opinion this ms can be accepted for its publication in Insects after major changes, the following are some general and specific suggestions that should be considered:

P1, L1-17, you have mentioned only “spinosad”, however, one more insecticide “cyromazine” is tested in this study, mention results of both insecticides in Abstract of ms

P2, L20, provide common name of insect and provide complete taxonomy of each italic name on first use in ms

P3, L88, can you provide a reference for your claim “onion production for multiple years”?

P3, L96, how the concentration of EPNs was calculated?

P3, L108-111, provide details of these EPN species, explain from where they have been isolated and had they been identified on molecular basis – acknowledge the source of EPNs?

P3, L112, what is 2000, if they are IJs then either in cm or ml, explain?

P3, L114, why you used double concentration of insecticides recommended in the field?

P3, L131, replace “scope” with “microscope or stereoscope”

P4, L176, use complete genus name of each italic name at start of paragraph/sentence throughout ms?

P4, L177, “ijs” should be in upper-case letters throughout

P5, L202-205, there is no information about seed treatment in Abstract

P6, L212-223, pl clarify, did you normalize the data before analysis? give reference for “Tukey-Kramer HSD test and rest of models and software etc.” and how the mortality was corrected, give reference?

P6, L244-276, provide total df value?

P6, L245, better to use p value as “<0.01” instead of “0.0001 and 0.001”

P9, L303, pl provide possible reasoning of this statement “negative impact on entomopathogenic nematode performance” and support with published literature

P10, L343-344, delete “For research articles with several authors, a short paragraph specifying their individual contributions must be provided. The following statements should be used”

The Discussion section should be more strengthened – the authors should provide the findings of few more relevant studies and then compare them logically with possible reasoning of variation in the other’s findings with the results of the current study

It is suggested authors should carefully check all the citation numbers with corresponding list of references, I noticed discrepancies in format/style of references and there is need to correct them following the format style of journal

Minor spell check is required

Round 2

Reviewer 2 Report

Some questions have not been properly addressed.

IPM/holistic sustainable agronomic practices issue: I think that the authors have not considered that researchers  should not follow any practice because there is an economic pressure or standardized (very dangerous) habits that human beings do not like to change because their brains like to keep in the comfort position. Researchers should show how to change the practices to achieve specific and general advantages; showing that an Integrated/holistic approach is sustainable and more profitable that “continuous crop” systems, expecially when you give a value to the side effects of chemical pesticide use and to the loss of biodiversity and ecosystem services.

This is a main responsability of the researchers.

In order to close, I invite the suthors to improve as follows:  

1)      Please  discuss  the insights above in the Introduction and in the Discussion; please show that different true IPM scenarios should be implemented and/or studied with some new references or, if you do not agree, try to demonstrate that  the unsustainable  (at least apparently they look  unsustainable) agronomic practices implemented are really the best solution (all considered, including economic aspects)  and do not cause the side effects I mention above.

2)      All the actual agronomic information should be given: which were the previous crops of the fields? Continuous onion cultivation as stated in the introduction?

Other  agronomic data are still missing;

3)      EPN are very susceptible to soil moisture levels; the soil moisture pattern over the experimental patterns should be given and discussed;

Sorry, no; you cannot ask the reader to look for key data needed to understand and replicate the experiment; in addition, general data of meteorological station somewhere close the experimental fields are not suitable for this kind of experiment; it is necessary to give the real data of the the field during the experiment; e.g. during the days of and  just after the EON application soil moisture …..;  irrigation water was supplied …… (mm….) …using…. (Table….).

No doubts about the soil and climatic conditions during the experiment should remain;

4)      Figure 1 in M&M presents Results and should be moved to Result section.

Reviewer 3 Report

I have carefully gone through the revised version of Filgueiras et al., there is no doubt on the hard efforts in revising this article - this study is providing novel data and would be a good addition in entomology literature - based on the importance of study I am convinced to accept this article for its publication in Insects in its current form

I believe minor grammar/spelling editing and uniform style of list of references will be done before its publication

Author Response

I have carefully gone through the revised version of Filgueiras et al., there is no doubt on the hard efforts in revising this article - this study is providing novel data and would be a good addition in entomology literature - based on the importance of study I am convinced to accept this article for its publication in Insects in its current form

Thank you for your feedback and suggestions on this paper.

Comments on the Quality of English Language:I believe minor grammar/spelling editing and uniform style of list of references will be done before its publication

That is correct. We are happy to work with the editorial team to polish the paper
and ensure uniform style before publication.